# Impaired semen quality, an increase of sperm morphological defects and DNA fragmentation associated with environmental pollution in urban population of young men from Western Siberia, Russia

**Maxim Kleshchev\*[ORCID], Alexander Osadchuk, Ludmila Osadchuk**

Department of Human Molecular Genetic, Federal Research Center 'Institute of Cytology and Genetics', the Siberian Branch of the Russian Academy of Sciences, Novosibirsk, Russia

☯ These authors contributed equally to this work.

\* max82cll@bionet.nsc.ru

## Abstract

Poor sperm morphology and an elevated DNA fragmentation level are considered to be related to spermiogenesis malfunctions as a result of genetic mutations and effects of environmental factors, including industrial pollution. Standardized cross-sectional population studies of sperm morphology defects and sperm DNA fragmentation, especially in regions with increased environmental pollution may be helpful to investigate an influence of industrial pollution and other population-related factors on spermiogenesis process. The aim of present study was to estimate an influence industrial pollution on sperm morphogenesis and sperm DNA fragmentation in men from the general population of the Western Siberia. The Novosibirsk and Kemerovo cities are located to same climatic conditions in Western Siberia but the Kemerovo city is characterized by increased environmental pollution especially by particulate matter (PM). The male volunteers living in Novosibirsk (n = 278) and Kemerovo (n = 258) were enrolled. Percentages of sperm morphological defects are counted after staining native ejaculate smears by Diff-Quick kits. DNA fragmentation was estimated by a SCSA technique. The residents of Kemerovo were characterized by lowered sperm count and sperm motility, elevated DNA fragmentation, poor sperm morphology and increased incidence of morphological effects of head (pyriform, elongated, round, abnormal acrosome and vacuolated chromatine), asymmetrical neck insertion and excess residual cytoplasm. Moreover, elevated DNA fragmentation was associated with lowered sperm count, sperm motility and increased percentages of several sperm morphology defects, with the place of residence affecting the relationships between conventional semen parameters, sperm morphology and DNA fragmentations. Our study suggests that excessive sperm head elongation and impaired acrosome formation can contribute to sperm morphology deterioration in men from polluted areas. Regional features in the relationships between sperm morphology, sperm count and DNA fragmentation were shown, suggesting an importance of studying sperm morphology pattern in men from different regions.

**Data Availability Statement:** All relevant data are within the manuscript and its Supporting Information files.

**Funding:** The study supervision, the data analysis and manuscript writing were supported by project nr19-15-00075 from the Russian Science Foundation.

**Competing interests:** The authors have declared that no competing interests exist.

## Introduction

Infertility is a complex reproductive disorder affecting nearly 15% couples, with 50% of cases are caused by male factor [1]. Sperm count, sperm motility and sperm morphology are considered to be conventional markers of male fertility [2], however a routine microscopic examination of spermatozoa does not always predict male fertility exactly and point to the reasons of infertility [3, 4]. In recent decades several additional tools have been developed for estimating a male fertility status, including evaluation of DNA fragmentation in spermatozoa. The numerous evidences show that DNA fragmentation is a good predictor of male fertility [5–7].

It should be noted that in many cases decreasing sperm parameters, especially sperm motility, morphology and DNA integrity might be caused by spermiogenesis impairments. Spermiogenesis is the complex process of morphological and biochemical differentiation of the haploid spermatid and involves the formation of acrosome, reshaping of the nuclear morphology, assembly of the flagellum and the structures responsible for sperm motility, chromatin condensation, elimination of residual cytoplasm [8–14]. Chromatin reorganization during spermiogenesis involves histone replacement by transition proteins and finally protamines. Moreover, physiological breaks in sperm DNA resulting from histone to protamine replacement are repaired [15]. These processes result in the formation of a species-specific sperm shape and a tightly packed chromatin resistant to an action of environmental agents, especially reactive oxygen species (ROS). Protamination failures cause DNA fragmentation because sperm fail to repair the physiological DNA breaks [15, 16] and incomplete protamination results in increased sperm vulnerability to ROS. Furthermore, excess residual cytoplasm (ERC) was shown to be the significant source of ROS [17, 18]. ROS is known to be one of the reasons causing DNA damage and lowered sperm quality and male fertility [15, 19]. Consequently, spermiogenesis deficiency may affects sperm morphology due to failing sperm resharping, result in elevated sperm DNA fragmentation due to impaired chromatin condensation as well as decreased sperm motility due to defective tail assembly.

The spermiogenesis errors result in an appearance in the ejaculate of sperm morphological defects presented in various proportions. To date several classification systems of sperm defects have been made which considered the structural features of head, midpiece and tail [2, 20, 21]. Several authors consider that the sperm defects rates in the ejaculate (sperm morphological pattern) are specific for the certain men [22]. The morphological pattern is believed to be caused by joint influences of genetic and external factors [21, 22], with sperm morphology being related to sperm motility [23] and DNA fragmentation level [24–28]. Sperm morphology pattern estimation is considered to be more informative for male reproductive assessment in comparison with simple counting sperm with normal morphology because high frequencies of some crucial abnormalities may reflect the changes to the spermiogenic processes with the functional consequences resulting in decreased fertility [21, 22, 29]. On the other hand, several studies reported lack the association between sperm morphology, DNA fragmentation [30, 31] and male fertility [32–34]. These discrepancies are usually considered to be caused by the methodological severities of sperm morphology assessment [35, 36], although it may evidence that the relationships between sperm parameters and sperm morphology pattern are complex, reflects sophistication of spermiogenesis process and depends on the interactions of numerous external factors. This circumstance suggests necessity of standardized detailed studying genetic and environmental factors affecting sperm morphology pattern and its relationships with other sperm parameters.

Spermiogenesis is sensitive to environmental conditions and lifestyle factors. Obesity [37, 38], smoking [39], andrological disorders [40] and chronic system illness [41] affect sperm morphology. Furthermore, environmental pollution is the severe factor affecting sperm count,

sperm motility, percentage of morphologically normal sperm [42], sperm morphological pattern [43] and DNA fragmentation level [44, 45]. Environmental pollutants include polycyclic hydrocarbons [46], heavy metals, gases, and particulate matter (PM) being components of industrial pollution [45, 47].

It should be noted that air and soil pollutions are related to geographic location and contribute to regional differences for sperm count, progressive motility and percentage morphologically normal sperm revealed by numerous studies [48–52].

Susceptibility of spermiogenesis to the environmental factors affecting human population in different regions and the importance of sperm morphology for male fertility assessment and understanding reasons and mechanisms of spermiogenesis disturbances suggest necessity of standardized cross-sectional population studies of sperm morphology defects and sperm DNA fragmentation, especially in regions with increased environmental pollution to investigate influence of industrial pollution and other population-related factors on spermiogenesis process. However, the population studies of sperm morphological pattern and sperm DNA damage are rarely [20, 53, 54]. To our knowledge, population studies including simultaneous detailed assessment sperm morphology pattern and DNA fragmentation in regions with different levels of industrial pollution has not been conducted. For human populations of Western Siberia cross-sectional study of sperm morphological pattern and DNA fragmentation has not been performed also, although certain areas in this territory are characterized by severe environmental pollution due to metallurgical, chemical and mining enterprises. In particular, the Novosibirsk and Kemerovo cities are located to same climatic conditions but Kemerovsky region (Kuzbass) and the Kemerovo city are characterized by increased environmental pollution especially by PM due to coal mining and the numerous metallurgical and chemical industries [55, 56]. In previous study it was shown that men from general populations of the Kemerovo city were characterized by lower sperm count, sperm motility, testosterone and inhibin B levels in comparison with the residents of Novosibirsk city [56]. Investigation of sperm morphology pattern and DNA fragmentation in this region may be interesting for evaluating complex industrial pollution on sperm morphogenesis and chromatin compaction.

The aim of the study was to estimate the influence of industrial pollution on sperm morphology pattern and sperm DNA fragmentation in men from the general populations of Western Siberia. To complete the picture, we discussed the above-mentioned previously published data on basic semen parameters (sperm count, concentration and motility, percentage of spermatozoa with normal morphology) together with our data on sperm morphological defects and DNA fragmentation in men from the general populations living in Novosibirsk and Kemerovo.

## Materials and methods

### Study population

The study in Novosibirsk covered period: October2011–May 2014, in Kemerovo October 2009–March 2013.

The recruitment protocol and physical examination methods had been described earlier in detail [56]. Briefly, male volunteers being interested in their reproductive health, with unknown fertility living in Novosibirsk (n = 278) and Kemerovo (n = 258) at least 5 years were enrolled in the study. The investigation was performed in 2010–2014 years. Inclusion criteria for participation in the study were absence of acute general diseases or chronic illness in an acute phase, and genital tract infections. All participants gave written informed consent to participate in the examination.

Each participant was examined by the some experienced andrologist to determine possible andrological disorders and to measure body weight, height, testicular volume and estimate secondary sexual characteristics. For all participants body mass index (BMI, kg/m$^2$) was calculated. To estimate possible effects of obesity on sperm morphology each participant was assigned to on one of three groups according to his BMI: normal weight (18.5–24.9 kg/m$^2$) overweight (25–29.9 kg/m$^2$) and obesity (BMI$\geq$30 kg/m$^2$).

Each participant filled in a standardized questionnaire including information on current age, place of birth, family status, previous or current urological diseases tobacco smoking, alcohol consumption and a history of fertility as well as the self-identified nationality of the volunteer and his parents.

The ethics committee of the Federal Research Center 'Institute of Cytology and Genetics', the Siberian Branch of the Russian Academy of Sciences approved the study.

## Semen samples

The semen samples were collected by masturbation and analyzed for semen volume (ml), sperm concentration (mln/ml), progressive motility (percentage) and sperm morphology. Semen volume, sperm concentration and sperm count and progressive motility for all subjects were estimated according to the "WHO laboratory manual for the examination and processing of human semen" [2] as described earlier [56, 57]. To study the relationships between impaired semen parameters (sperm concentration and sperm motility) and percentages of sperm morphological defects each participant was assigned to semen quality status (SQS) according to the sperm quality thresholds provided by the WHO [2]: normozoospermia (sperm concentration and percentage of progressively motile spermatozoa equal to or above the lower reference limits) and pathozoospermia (sperm concentration < 15 mln/ml and/or percentage of progressively motile spermatozoa<32%).

## Sperm morphology analysis

To assess sperm morphology, ejaculate smears were prepared, fixed by methanol (during 1 min) and stained using Diff-Quick kits (Abris plus, Russia) according to the "WHO manual. . ." [2]. Spermatozoa were examined for morphology using the bright-field optical microscope (Carl Zeiss, Germany) at ×1000 magnification with oil-immersion. The microscope was equipped by the digital camera AxioScope with a special software (AxioVision 9.0) enabling to perform sperm morphometry. Two hundred sperm for each semen sample were assessed. Percentage of normal sperm was estimated twice in random and blinded order by two experienced investigators. Spearman correlation coefficient between between percentages sperm with normal morphology obtained by first and second count was 0.78, statistically significant difference was not observed. The differences between first and second counts of normal sperm morphology were acceptable according criteria provided by «WHO laboratory manual. . .» [2] for each sample. The mean value for percentage of normal sperm was calculated.

Percentages of sperm morphological defects were estimated in random and blinded order by the only experienced researcher (M.K.).

## Sperm morphology classification

The sperm were classified as normal, according to the criteria for normal sperm morphology provided by "WHO laboratory manual. . ." [2]. Other sperm referred to as abnormal, including borderline forms.

The classification of sperm defects was performed according to the classification scheme proposed by [2]. Unfortunately, the manual does not provide exact definitions of each sperm morphology abnormality but the photos of some sperm defects are presented only.

Consequently, additional criteria for definitions of each sperm defects category were used to standardize sperm morphology classification in our study.

A total of 14 types of anomalies were taken into account.

1. **Amorphous head**. Amorphous heads include heads vastly misshaped, with irregular edges, asymmetric, with an expanded postacrosomal zone, but not referred to pear-shaped or elongated heads.

2. **Pyriform head**. This type includes heads with severely elongated and narrowed postacrosomal zone [22]. The ratio of head length /width was $\geq 2$, with width being measured at equatorial zone [58].

3. **Elongated head**. The head has length /width ratio >2 [58], but unlike pyriform head width of acrosome and postacrosome zone are the same.

4. **Round head**. The head has length /width ratio = 1 [58].

5. **Acrosome defect**. The acrosome area is less than 40% or greater than 70% of the head area [2].

6. **Vacuolated head.** The head has more than two vacuoles occupying more than 20% of the head area or located in the postacrosomal region [2].

7**. Thick midpiece**. The midpiece width is more than one mkm [2].

8. **Thin midpiece**. The midpiece width is less than the principal piece width [58].

9. **Bent neck**. The midpiece forms an angle of 90˚ (or less) to the long axis of the sperm head [58].

10. **Excess residual cytoplasm (ERC).** The large (more than 1/3 of the head area) cytoplasmic remnant is presented [2].

11. **Asymmetrical neck insertion**. The attachment of the midpiece is not aligned with central axis of the head [2].

13. **Coiled tail.** The tail is coiled itself (> 360˚) [2].

14. **Short tail.** The tail length is less than 45 mkm [2].

Defects in each spermatozoon were recorded using laboratory counter in multiple-entry system. A percentage of each sperm abnormality and teratozoospermia index (TZI) was calculated according "WHO laboratory manual. . ." [2].

## DNA fragmentation estimation

Sperm chromatin structure assay (SCSA) has been used to estimate DNA fragmentation level [7, 59]. Immediately after ejaculate collection, semen aliquot (300 mkl) was frozen and stores (-40˚C) until analysis. To analyze sperm DNA fragmentation, the sample diluted in TNE-buffer (0.01 M Tris, 1 mM EDTA, 0.15 MNaCl, pH—7.4) to the sperm concentration 1 million / ml and then 200 μl of acid buffer (0.1% Triton-X-100, 0.15 M NaCl, 0.08 N HCl, pH -1,2). was added to 100 mkl diluted ejaculate. After incubation for 30 s, 600 μl of dye solution containing 6 mg/L acridine orange, 0.2 M $Na_2HPO_4$, 1 mM EDTA (disodium), 0.15 M NaCl, 0.1 M citric acid (pH 6.0) was added. Spermatozoa with red and green fluorescence were counted by fluorescence cytometer Guava Easy CyteMini ("Guava", USA). Each sample was evaluated thrice (5000 cells in each estimation). DNA fragmentation index (DFI) was calculated as percentage of cells with red fluorescence (sperm with fragmented DNA) of the total number of cells with red and green fluorescence.

According to the DFI thresholds proposed by some authors [7, 59], each participant was assigned to one of three DNA fragmentation levels: normal (DFI<15%), slightly increased (15% $\leq$ DFI < 27%) and severe increased (DFI $\geq$ 27%). To analyze the relationships between sperm DNA fragmentation and sperm morphology slightly increased and severe increased groups were combined.

## Statistical analysis

The statistical analysis of the data was performed using the statistical package "Statistica" (version 8.0). Sperm quality parameters, DFI and percentages of sperm morphological defects were not normally distributed according to the Kolmogorov–Smirnov test. The data were best transformed by a square root transformation for sperm count and sperm concentration; a log transformation for TZI and an arcsine transformation for DFI, percentages of progressively motile spermatozoa, morphologically normal spermatozoa and all sperm morphology defects. Descriptive statistics in the tables, on the figures and in the text are presented using untransformed data as mean and its standard deviation and median with 5th - 95th percentiles. Spearman's correlation coefficients were used to determine the relationships among all studied parameters. To investigate influences of region and semen quality (or DFI level) on the percentages of sperm morphology defects a two-way analysis of covariance (ANCOVA) was used. Categorical predictors (factors) in ANCOVA were the place of residence and SQS (lowered or normal) or DFI level (normal or increased), with age and abstinence time being considered as covariates. Duncan's test was used to determine statistical significance of the differences between the groups. A p value <0.05 was regarded as statistically significant.

## Results

### Anthropometrical and sociodemographic characteristic

The demographic and morphometric characteristics of the studied populations are present in Table 1. As shown in the Table 1 men from Kemerovo city were characterized by higher age and bitesticular volume but lower abstinence time in comparison with participants from Novosibirsk city. No statistically significant differences were revealed for proportions of cigarette smokers, alcohol drinkers and men with overweight and obesity. Proportions of men suffered from varicocele (grade I and grade II) were higher in Kemerovo city compared to those from Novosibirsk city. Percentages of the subjects with prostatitis and testicular cysts were higher in Novosibirsk city compared to those from Kemerovo city.

**Climate and pollution level in Novosibirsk and Kemerovo.**   The data on air temperature and pollution levels for Novosibirsk and Kemerovo in 2009–2014 years (period of the investigation) are shown in Table 2.

Kemerovo city was characterized by severe increased emissions volume in atmospheric air of solid particles, sulfur dioxide, nitrogen oxides and carbon monoxide in comparison with Novosibirsk city.

### Sperm morphology: Regional differences

Of the 561 participants, some men (n = 16) refused to donate ejaculate, azoospermic men (n = 9) has been excluded, so 536 semen samples (278 samples from Novosibirsk and 258 it from Kemerovo) were assessed. DFI and percentages of normal sperm and sperm morphological defects are presented in Table 3.

Significant effects of the residence place and SQS was showed by ANCOVA for all semen parameters and percentages of sperm morphological defects. The results of ANCOVA (categorical factors–the residence place and SQS; covariates–age and abstinence time) are presented in S1 Table.

The residents of Kemerovo city had significantly lower percentages of morphologically normal sperm and amorphous heads and higher TZI, incidences of pyriform, elongated, round, vacuolated heads, sperm with abnormal acrosome, spermatozoa with thin, thick midpiece, asymmetrical neck insertion and ERC compared to men from Novosibirsk (Table 3).

**Table 1. Demographic and anthropometric characteristics in men of different regions.**

| Characteristics | Novosibirsk (n = 280) | Kemerovo (n = 258) | P value (between groups) |
|---|---|---|---|
| Age (day), mean(SD) | 22,25(3,51) | 23,76(5,04) | 0,0003[a] |
| Body weight (kg), mean(SD) | 75,87(14,05) | 75,97(12,20) | 0,5791[a] |
| Height (cm), mean(SD) | 179,49(6,90) | 178,68(6,84) | 0,1763[b] |
| BMI(kg/m$^2$), mean(SD) | 23,50(3,85) | 23,76(3,26) | 0,1223[a] |
| BTV, ml mean(SD) | 36,64(8,44) | 41,19(7,86) | <0,0001[a] |
| Abstinence, day mean(SD) | 5,60(5,61) | 4,30(5,58) | <0,0001[a] |
| Cigarette smokers n(%) | 70(25.00) | 83(36,38) | 0.0500[c] |
| Alcohol drinkers, n(%) | 209(74,91) | 193(75,98) | 0.5765 [c] |
| Obesity, n(%) | | | 0,8437 [c] |
| Normal weight | 197(70.86) | 180(70.59) | |
| Overweight | 64(23.02) | 62(24.31) | |
| Obesity | 17(6.12) | 13(5.10) | |
| Current andrology disease, n(%) | 97(34.89) | 82(32.41) | 0.0022[c] |
| Varicocele, grade I | 3(1.08) | 14(5.53) | |
| Varicocele, grade II | 6(2.16) | 15(5.93) | |
| Varicocele, grade III | 5(1.8) | 4(1.58) | |
| Testicular cysts | 26(9.35) | 12(4.74) | |
| Prostatitis | 36(12.95) | 17(6.72) | |
| Epididymitis | 6(2.16) | 5(1.98) | |
| Other | 15(5.4) | 15(5.93) | |

*Note.* Results based on raw data.

[a]- Mann–Whitney test was used to compare anthropometric and demographic parameters.

[b]–t-test was used to compare height of participants.

[c]—Chi-square test was performed to compare the frequency parameters.

Abbreviations: SD—standard deviation; (5–95) - 5th–95th percentile; BMI–body mass index; BTV–bitesticular volume (paired testicular volume).

**Table 2. Climatic and pollution parameters in two different regions.**

| Parameters | Novosibirsk | Kemerovo |
|---|---|---|
| **Emissions volume of air pollutants (annual average), thousand tons, mean (SD)[a]** | | |
| Solid particles | 47.67(3.25) | 150.31(13.76) |
| SO$_2$ | 45.93(5.04) | 109.41(10.75) |
| CO | 277.31(12.61) | 363.86(110.36) |
| NOx | 74.65(3.34) | 88.85(34.55) |
| **Average air temperature, °C[b]** | | |
| January | -17,5 | -17,0 |
| Jule | 19,4 | 19,0 |

Note.

[a] The data on emissions volume of pollutants (annual average) for Novosibirsk in 2011–2014 years and Kemerovo in 2009–2013 years were obtained from public government reports about environment status and conservation of Kemerovo [available from: http://kuzbasseco.ru/doklady/o-sostoyanii-okruzhayushhej-sredy-kemerovskoj-oblasti/] and Novosibirsk [available from: https://www.nso.ru/page/2624] regions. The present time period corresponds to time of the study.

[b] The data on air temperature were obtained from Wikipedia.org for Novosibirsk [https://en.wikipedia.org/wiki/Novosibirsk] and Kemerovo [https://en.wikipedia.org/wiki/Kemerovo].

**Table 3. DNA fragmentation and percentages of sperm morphology defects in men of different regions.**

| Parameters | Novosibirsk(n = 280) | | Kemerovo (n = 255) | | |
|---|---|---|---|---|---|
| | Mean(SD) | Median(5–95) | Mean(SD) | Median(5–95) | P value |
| DFI, % | **8.23(6.56)** | 6.20(2.59–23.17) | **11.68(8.63)** | 8.89(2.99–31.12) | **0.0025** |
| Normal sperm, % | **7.94(3.11)** | 8.00(2.75–13.75) | **6.42(3.01)** | 6.50(1.48–11.25) | **<0.0001** |
| TZI | **1.45(0.11)** | 1.44(1.3–1.65) | **1.53(0.13)** | 1.51(1.34–1.79) | **<0.0001** |
| **Head abnormalities** | | | | | |
| Amorphous, % | **69.20(12.48)** | 71.50(45.00–85.75) | **56.22(13.98)** | 58.00(29.50–76.00) | **<0.0001** |
| Pyriform, % | **7.21(8.19)** | 4.50(0.50–24.25) | **12.47(11.38)** | 9.00(1.00–38.50) | **<0.0001** |
| Elongated, % | **8.47(7.63)** | 6.00(1.00–24.75) | **14.31(10.09)** | 11.50(2.00–34.50) | **<0.0001** |
| Round, % | **1.13(1.60)** | 0.50(0.00–4.00) | **2.26(2.48)** | 1.50(0.00–7.50) | **<0.0001** |
| Large, % | 0.14(0.29) | 0.00(0.00–0.50) | 0.13(0.31) | 0.00(0.00–0.50) | 0.3552 |
| Small, % | **0.63(0.94)** | 0.50(0.00–2.50) | **0.47(0.69)** | 0.00(0.00–2.00) | **0.0191** |
| Double, % | 0.06(0.22) | 0.00(0.00–0.50) | 0.04(0.17) | 0.00(0.00–0.50) | 0.1187 |
| Vacuolated, % | **9.44(6.29)** | 8.00(2.00–22.25) | 12.44(7.39) | 12.00(3.00–24.50) | **<0.0001** |
| Abnormal acrosome, % | **15.86(9.73)** | 14.00(5.50–34.00) | **20.09(12.35)** | 17.00(5.50–44.50) | **<0.0001** |
| **Midpiece abnormalities** | | | | | |
| Bent_head, % | 5.75(3.99) | 4.50(1.00–14.50) | 5.97(4.27) | 5.00(1.50–13.50) | 0.4023 |
| Asymmetrical neck insertion, % | **12.91(5.47)** | 12.50(5.00–22.75) | **24.22(6.24)** | 24.00(13.00–34.50) | **<0.0001** |
| Thick mipiece, % | **6.28(3.63)** | 5.50(1.50–13.50) | **7.35(3.75)** | 7.00(2.50–14.00) | **0.0005** |
| Thin midpiece, % | **0.91(1.16)** | 0.50(0.00–2.50) | **1.18(1.28)** | 1.00(0.00–3.50) | **0.0051** |
| **Tail abnormalities** | | | | | |
| Double tail, % | 1.27(1.07) | 1.00(0.00–3.13) | 1.34(1.33) | 1.00(0.00–4.00) | 0.9941 |
| Coiled tail,% | 10.05(5.45) | 9.00(3.50–20.75) | 10.6(6.35) | 9.00(3.00–23.00) | 0.3573 |
| Short tail, % | 2.62(2.23) | 2.00(0.00–6.50) | 2.46(2.26) | 2.00(0.00–7.00) | 0.4412 |
| **Excess residual cytoplasm** | | | | | |
| ERC, % | **7.01(4.50)** | 6.00(1.50–15.00) | **8.58(4.7)** | 8.00(2.50–16.00) | **0.0002** |
| **Abnormalities in different parts of spermatozoon** | | | | | |
| Head, % | **51.06(8.55)** | 52.00(36.50–64.25) | **40.68(8.54)** | 40.50(26.47–55.00) | **<0.0001** |
| Midpiece,% | **3.12(2.35)** | 2.50(0.50–8.00) | **5.10(3.05)** | 5.00(0.50–10.00) | **<0.0001** |
| Tail, % | **1.60(1.60)** | 1.00(0.00–5.00) | 1.04(1.09) | 0.50(0.00–3.50) | **<0.0001** |
| Head&Midpiece_% | **24.08(6.89)** | 23.00(14.25–37.00) | **33.55(7.44)** | 33.50(21.50–47.00) | **<0.0001** |
| Head&Tail_% | 9.07(4.63) | 8.00(3.00–18.25) | 9.29(5.53) | 8.00(3.00–19.50) | 0.6485 |
| Midpiece&Tail_% | **0.21(0.37)** | 0.00(0.00–1.00) | 0.37(0.54) | **0.00(0.00–1.50)** | **0.0012** |
| Head&Midpiece&Tail_% | **3.01(2.27)** | 2.50(0.50–7.00) | **3.68(2.75)** | 3.00(0.50–9.00) | **0.0066** |

*Note*. Results based on raw data. Analysis of variance used to compare all parameters. Significant (p<0.05) differences between groups are highlighted by bold text.
Abbreviations: SD—standard deviation; (5–95) - 5th–95th percentile; DFI–DNA fragmentation index; TZI–teratozoospermia index; ERC–excess residual cytoplasm.

The subjects with lowered sperm concentration and/or motility were characterized by higher percentages of all sperm defects except large heads (S2 Table). Moreover, interactions of the residence place and SQS were shown for percentages of amorphous, pyriform, vacuolated heads, ERC and asymmetrical tail insertion. The residents of Kemerovo with lowered semen quality were characterized by more higher percentages of pyriform, vacuolated heads and ERC compared to men with normal semen quality (Fig 1), while for normal and pathozoospermic men from Novosibirsk no differences in percentages of this defects were found. However, the residents of Novosibirsk with lowered sperm quality were characterized by higher percentages of amorphous head and asymmetrical tail insertions compared to men with normal sperm quality, but these differences was not showed for the residents of Kemerovo.

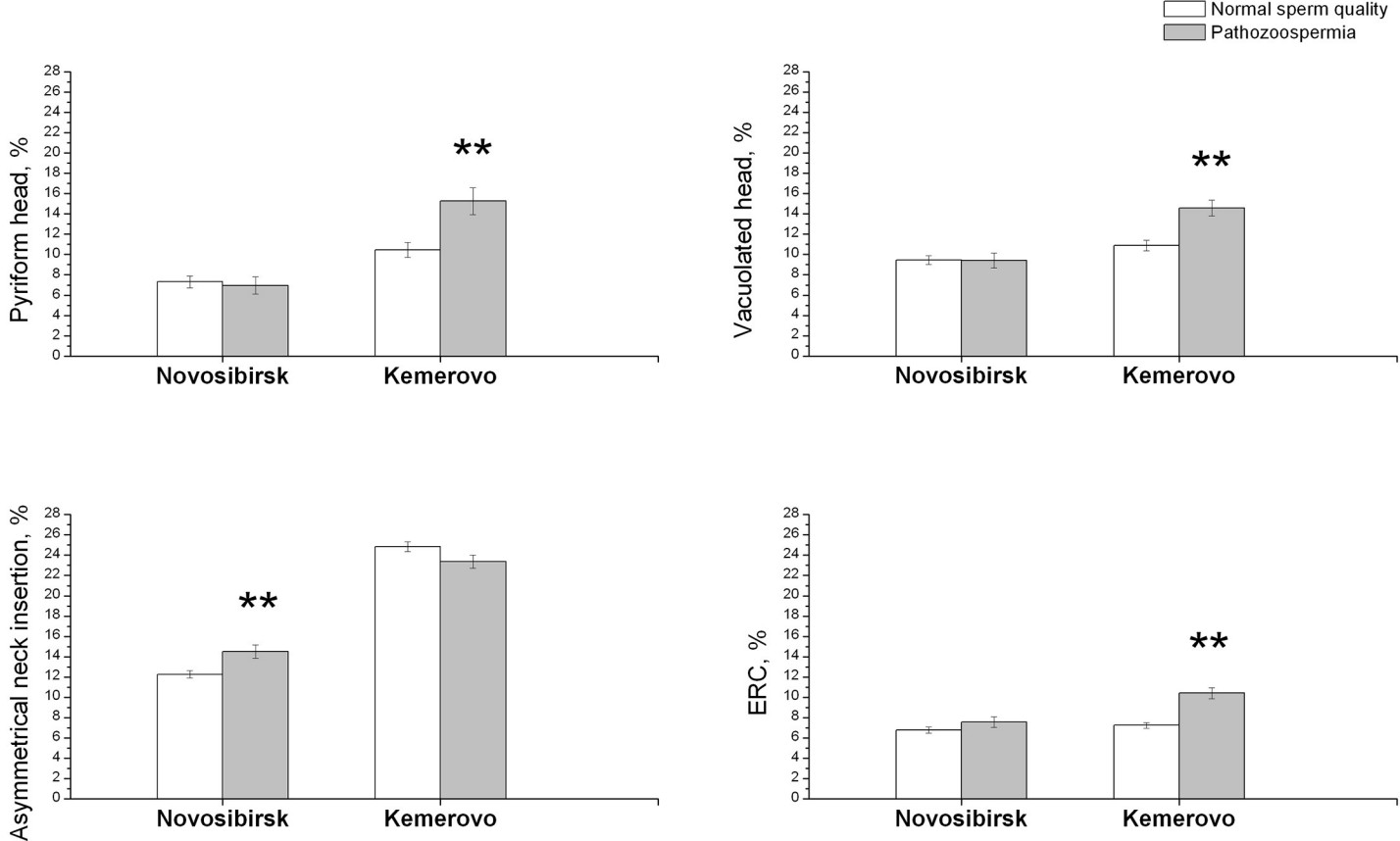

**Fig 1. Percentages of several sperm morphological defects in men from Novosibirsk and Kemerovo cities with pathozoospermia and normal semen quality.** ** — significance of differences between groups (p<0.01). Abbreviations: ERC—excess residual cytoplasm.

## DNA fragmentation level: Regional differences and the relationships with sperm morphology

Men from Kemerovo had higher DFI in comparison with the subjects from Novosibirsk (Table 3). Among the participants from Kemerovo 78.85% of the subjects were characterized by normal DFI, whereas other had increased (12.5%) and severe increased (8.65%) DFI. Among the men from Novosibirsk 89.33% of the subjects had normal DFI, while other had increased (8.00%) and severe increased (2.67%) DFI. The differences in proportions of subjects with normal and increased DFI were significant according to the Chi-square test ($\chi 2 = 6.41$, p = 0.0405).

The ANCOVA using the residence place and DNA fragmentation level as categorical factors showed significant influence of DNA fragmentation level on sperm count, sperm concentration, progressive sperm motility, as well as percentages of sperm with normal morphology, TZI and incidences of several sperm morphology defects (S3 Table). The subjects with increased DFI had lower sperm count, sperm concentration, progressive sperm motility, percentage of sperm with normal morphology but higher TZI, percentages of elongated and double head, abnormal acrosome, bent head, asymmetrical neck insertion, and coiled tail (p<0.01), compared to men with normal DNA fragmentation level (Table 4).

The interactions of the residence place and DNA fragmentation status were shown by ANCOVA for sperm count, sperm concentrations (S3 Table). The residents of Novosibirsk with increased DFI were characterized by lower sperm count and sperm concentration

**Table 4. Sperm quality and sperm morphology defects in men with different DNA fragmentation level.**

| Parameters | DFI level | | | | P value |
|---|---|---|---|---|---|
| | Normal (n = 219) | | Increased (n = 38) | | |
| | Mean(SD) | Median(5–95) | Mean(SD) | Median(5–95) | |
| Sperm count, mln | **221.16 (144.08)** | 205.69 (29.58–467.17) | **167.85 (235.26)** | 83.68 (3.1–963.99) | **0.0001** |
| Sperm concentration, mln/ml | **60.44 (37.36)** | 53.63 (11.88–134.93) | **37.31 (38.41)** | 22.19 (1.05–129.50) | **<0.0001** |
| Progressive motility, % | **51.94(25.38)** | 53.53(5.90–92.37) | **23.76(21.71)** | 18.08(1.1–65.88) | **<0.0001** |
| Normal sperm, % | **7.86(3.02)** | 7.75(2.50–13.50) | **5.32(3.01)** | 4.13(1.06–10.75) | **<0.0001** |
| TZI | **1.47(0.12)** | 1.46(1.30–1.70) | **1.56(0.16)** | 1.49(1.35–1.88) | **<0.0001** |
| **Head abnormalities** | | | | | |
| Amorphous, % | 63.48(13.84) | 66.50(36.50–83.00) | 64.28(15.45) | 66.75(29.5–85.5) | 0.6829 |
| Pyriform, % | 9.00(9.60) | 6.50(0.50–30.00) | 8.63(7.81) | 6.00(1.00–24.50) | 0.8670 |
| Elongated, % | **10.10(9.11)** | 7.00(1.00–28.50) | **13.58(10.71)** | 11.51(1.50–35.29) | **0.0151** |
| Round, % | 1.44(1.94) | 1.00(0.00–5.00) | 1.71(2.11) | 1.00(0.00–8.00) | 0.4364 |
| Large, % | 0.15(0.31) | 0.00(0.00–1.00) | 0.09(0.23) | 0.00(0.00–0.50) | 0.3032 |
| Small, % | 0.56(0.91) | 0.00(0.00–2.50) | 0.47(0.78) | 0.00(0.00–2.50) | 0.4698 |
| Double, % | 0.04(0.13) | 0.00(0.00–0.50) | 0.16(0.48) | 0.00(0.00–1.00) | **0.0167** |
| Vacuolated, % | 11.09(6.94) | 10.00(2.50–24.50) | 12.50(6.04) | 11.75(3.00–23.00) | 0.1475 |
| Abnormal acrosome, % | **17.05(9.71)** | 14.50(6.00–36.00) | **25.19(14.66)** | 21.00(6.00–53.00) | **<0.0001** |
| **Midpiece abnormalities** | | | | | |
| Bent_head, % | **6.27(4.00)** | 5.50(1.50–14.00) | **9.36(6.36)** | 8.00(2.00–19.00) | **0.0001** |
| Asymmetrical neck insertion, % | **16.27(8.46)** | 14.00(5.50–33.00) | **19.15(8.08)** | 18.75(5.50–34.50) | **0.0015** |
| Thick mipiece, % | 6.62(3.52) | 6.00(2.00–13.50) | 6.90(4.17) | 5.75(1.50–14.50) | 0.7441 |
| Thin midpiece, % | 0.97(1.13) | 0.50(0.00–3.00) | 1.37(1.84) | 1.00(0.00–4.00) | 0.1592 |
| **Tail abnormalities** | | | | | |
| Double tail, % | 1.29(1.17) | 1.00(0.00–3.50) | 1.53(1.71) | 1.00(0.00–6.50) | 0.7545 |
| Coiled tail,% | **9.87(5.61)** | 8.50(3.50–21.00) | **13.08(5.45)** | 12.25(5.00–23.50) | **0.0003** |
| Short tail, % | 2.43(2.18) | 2.00(0.00–6.50) | 3.47(2.92) | 3.00(0.00–11.50) | 0.5657 |
| **Excess residual cytoplasm** | | | | | |
| ERC, % | 7.38(4.24) | 6.50(2.50–15.50) | 7.59(4.04) | 6.50(2.50–14.50) | 0.7481 |
| **Abnormalities in different parts of spermatozoon** | | | | | |
| Head, % | **47.37(9.21)** | 47.00(32.50–63.00) | **41.60(12.70)** | 42.25(18.50–63.00) | **0.0001** |
| Midpiece,% | **4.39(2.67)** | 4.00(0.50–9.00) | **3.63(2.92)** | 3.00(0.00–9.50) | **0.0235** |
| Tail, % | 1.67(1.64) | 1.00(0.00–5.50) | 1.41(1.25) | 1.00(0.00–4.31) | 0.4548 |
| Head&Midpiece_% | **26.82(7.90)** | 26.50(15.50–40.00) | **31.85(9.52)** | 30.75(18.00–53.50) | **<0.0001** |
| Head&Tail_% | **8.54(4.61)** | 7.50(3.00–19.00) | **12.14(5.00)** | 11.50(4.00–21.00) | **<0.0001** |
| Midpiece&Tail_% | 0.3(0.49) | 0.00(0.00–1.50) | 0.25(0.42) | 0.00(0.00–1.00) | 0.449632 |
| Head&Midpiece&Tail_% | **3.04(2.45)** | 2.50(0.00–7.50) | **4.24(3.24)** | 3.50-(0.50–10.53) | **0.0134** |

*Note*. Results based on raw data. Analysis of variance used to compare all parameters. Significant (p<0.05) differences between groups are highlighted by bold text.
Abbreviations: SD—standard deviation; (5–95) - 5th–95th percentile; DFI–DNA fragmentation index; TZI–teratozoospermia index; ERC–excess residual cytoplasm.

compared to men with normal DFI (Fig 2). However, there are no differences for these parameters in the subjects from Kemerovo.

## The effects of lifestyle factors on sperm morphology and DNA fragmentation in Novosibirsk and Kemerovo

Significant effects of the smoking were showed by ANCOVA (categorical factors–the residence place and smoking; covariates–age and abstinence time) for semen parameters and percentages of several sperm morphological defects (S4 Table).

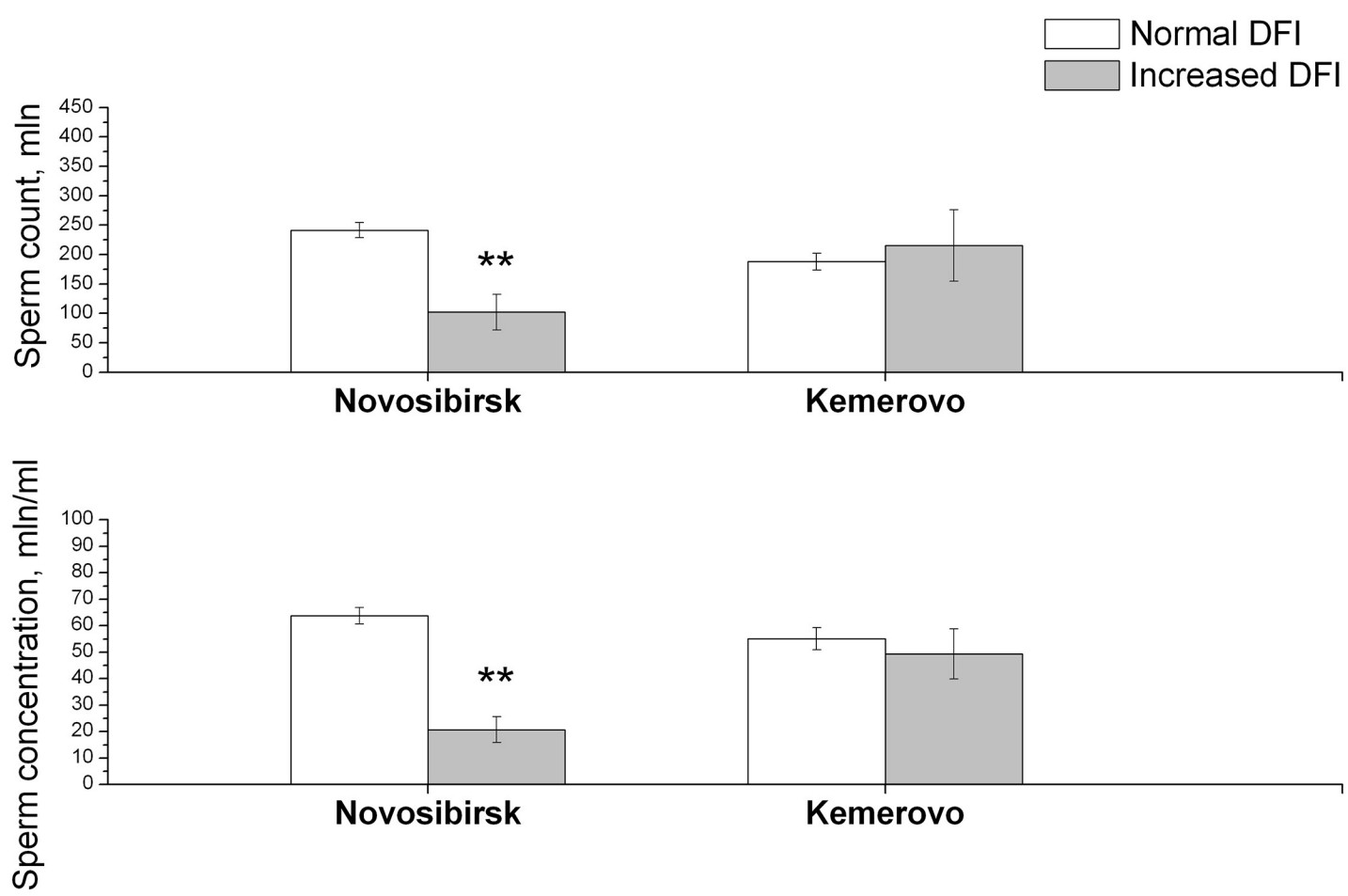

**Fig 2. Sperm count and sperm concentration in men from Novosibirsk and Kemerovo cities with normal and increased DFI.** \*\*—significance of differences between groups (p<0.01). Abbreviations: DFI—DNA fragmentation index.

Smokers were characterized by decreased sperm count, progressive motility as well as increased TZI, DFI, and percentages of elongated, double, vacuolated heads, spermatozoa with asymmetrical neck insertion and ERC, elevated incidence of combined defects of head and midpiece, combined abnormalities of head, midpiece and tail in comparison with non-smokers (Table 5).

Alcohol consumption slightly increased percentage of vacuolated heads but did not affect on percentages of other sperm defects as well as sperm quality and DNA fragmentation (The results are presented in S5 and S6 Tables).

No significant differences in sperm count, sperm concentration, progressive sperm motility, DFI, percentage of sperm with normal morphology and proportions of all sperm defects were found among men with normal body weight, overweight and obesity as shown in S7 and S8 Tables.

The subjects suffered from testicular cysts had slightly (p<0.05) decreased incidence of normal sperm compared to healthy men (mean±SD): 6.22±3.15% and 7.39±3.18% in sick and healthy men accordingly. No significant differences in sperm count, sperm motility, TZI, SDI and proportions of sperm morphological defects were not revealed between healthy men and the subjects with other andrological diseases (varicocele, prostatitis, epididymitis).

**Table 5. The effects of smoking on sperm quality and sperm morphology.**

| Parameters | Non-smokers (n = 381) | | Smokers (n = 153) | | P value |
|---|---|---|---|---|---|
| | Mean(SD) | Median (5–95) | Mean(SD) | Median (5–95) | |
| Sperm count, mln | **221.49(207.90)** | 181.94(26.09–467.17) | **189.68(164.15)** | 160.58(5.84–464.21) | **0.0016** |
| Sperm concentration, mln/ml | 57.43(41.54) | 48.91(7.38–134.25) | 55.03(41.94) | 44.13(3.38–142.55) | 0.0580 |
| Progressive motility, % | **46.59(26.03)** | 44.99(4.95–89.53) | **41.63(27.83)** | 40.45(1.97–86.13) | **0.0051** |
| Normal sperm, % | 7.32(3.14) | 7.25(2.50–13.00) | 6.96(3.18) | 7.00(1.75–11.75) | 0.1086 |
| TZI | **1.48(0.12)** | 1.47(1.32–1.70) | **1.51(0.14)** | 1.49(1.31–1.80) | **0.0007** |
| DFI, % | **8.61(5.99)** | 7.01(2.78–21.32) | **12.4(10.51)** | 8.13(2.69–34.79) | **0.0056** |
| **Head abnormalities** | | | | | |
| Amorphous, % | **63.8(14.79)** | 66.00(34.00–84.50) | **61.23(14.27)** | 62.50(34.00–80.75) | **0.0362** |
| Pyriform, % | 9.46(10.22) | 6.50(0.50–34.00) | 10.34(10.09) | 7.00(0.75–35.33) | 0.1577 |
| Elongated, % | **10.71(9.26)** | 8.00(1.00–28.00) | **12.43(9.23)** | 9.50(1.50–29.75) | **0.0089** |
| Round, % | 1.62(2.03) | 1.00(0.00–5.50) | 1.79(2.4) | 1.00(0.00–7.50) | 0.4229 |
| Large, % | 0.13(0.29) | 0.00(0.00–0.50) | 0.14(0.32) | 0.00(0.00–0.50) | 0.8403 |
| Small, % | 0.58(0.88) | 0.50(0.00–2.50) | 0.48(0.71) | 0.00(0.00–2.00) | 0.2797 |
| Double, % | **0.04(0.16)** | 0.00(0.00–0.50) | **0.07(0.27)** | 0.00(0.00–0.50) | **0.0463** |
| Vacuolated, % | **10.50(6.94)** | 9.00(2.25–22.50) | **11.77(7.08)** | 10.50(2.50–24.00) | **0.0247** |
| Abnormal acrosome, % | 17.31(10.37) | 15.00(6.00–38.00) | 19.10(12.98) | 15.50(5.00–47.00) | 0.0811 |
| **Midpiece abnormalities** | | | | | |
| Bent_head, % | 5.72(3.80) | 4.50(1.00–13.33) | 6.17(4.85) | 5.00(1.50–16.00) | 0.3206 |
| Asymmetrical neck insertion, % | **17.64(8.10)** | 16.50(6.00–31.50) | **19.94(8.04)** | 20.50(7.00–33.50) | **<0.0001** |
| Thick mipiece, % | 6.72(3.75) | 6.00(2.00–14.00) | 6.93(3.68) | 6.50(2.50–13.00) | 0.6052 |
| Thin midpiece, % | 0.98(1.08) | 0.50(0.00–3.00) | 1.17(1.52) | 1.00(0.00–3.50) | 0.2588 |
| **Tail abnormalities** | | | | | |
| Double tail, % | 1.32(1.20) | 1.00(0.00–3.50) | 1.26(1.20) | 1.00(0.00–3.50) | 0.8334 |
| Coiled tail,% | 10.15(6.01) | 9.00(3.00–21.50) | 10.73(5.62) | 9.50(3.50–22.50) | 0.3838 |
| Short tail, % | 2.44(2.10) | 2.00(0.50–6.00) | 2.79(2.55) | 2.00(0.00–8.82) | 0.2010 |
| **Excess residual cytoplasm** | | | | | |
| ERC, % | **7.56(4.54)** | 6.50(2.00–15.50) | **8.26(4.94)** | 7.50(2.50–17.00) | **0.0328** |
| **Abnormalities in different parts of spermatozoon** | | | | | |
| Head, % | **47.01(9.73)** | 47.00(30.50–62.75) | **43.88(10.35)** | 45.00(26.00–62.00) | **<0.0001** |
| Midpiece,% | 3.99(2.76) | 3.50(0.50–9.00) | 4.27(3.14) | 3.50(0.50–11.00) | 0.3938 |
| Tail, % | 1.38(1.47) | 1.00(0.00–4.00) | 1.22(1.23) | 1.00(0.00–3.50) | 0.1800 |
| Head&Midpiece_% | **27.9(8.44)** | 27.50(15.50–43.00) | **30.27(8.72)** | 30.00(17.00–44.00) | **<0.0001** |
| Head&Tail_% | 9.07(5.13) | 8.00(3.00–18.50) | 9.41(4.96) | 8.50(3.50–18.50) | 0.6890 |
| Midpiece&Tail_% | 0.28(0.43) | 0.00(0.00–1.00) | 0.32(0.53) | 0.00(0.00–1.50) | 0.9385 |
| Head&Midpiece&Tail_% | **3.14(2.36)** | 2.50(0.50–7.50) | **3.81(2.87)** | 3.00(0.50–9.75) | **<0.0001** |

*Note.* Results based on raw data. Analysis of variance used to compare all parameters. Significant (p<0.05) differences between groups are highlighted by bold text.
Abbreviations: SD—standard deviation; (5–95) - 5th–95th percentile; DFI–DNA fragmentation index; TZI–teratozoospermia index; ERC–excess residual cytoplasm.

## Correlations between DFI, sperm morphology and other semen parameters

Information on the correlation between DNA fragmentation and percentages of sperm morphology defects is provided in the S9 Table.

When considering the entire population, DFI negatively correlated with sperm count(-0,27, p<0.01), sperm concentration (-0,30, p<0.01), progressive motility (-0,46, p<0.01) and percentage of normal forms (-0,36, p<0.01). The significant positive correlations were revealed between DFI and TZI (0,38, p<0.01), percentages of elongated (0,23, p<0.01), round (0,19,

p<0.01) vacuolated heads (0,26, p<0.01), abnormal acrosome (0,35, p<0.01) asymmetrical neck insertion (0,27, p<0.01), coiled (0,21, p<0.01) and short tails (0,19, p<0.01).

Moreover, as shown in S9 Table, regional differences in the correlations between percentages of sperm morphology defects and DNA fragmentation were revealed. DNA fragmentation in men from Novosibirsk significantly correlated with sperm count, percentages of elongated heads, abnormal acrosome, asymmetrical neck insertion, combined defects in heads, midpiece and tails, whereas no significant correlations between these parameters were revealed for residents Kemerovo. On the other hand, DFI in men from Kemerovo significantly correlated with the frequency of ERC but no statistically significant correlation was shown for the residents of Novosibirsk.

As shown in S10 Table, sperm concentration negatively correlated with TZI (-0.39, p<0.01), percentages of acrosome defects (-0.29, p<0.01), bent head (-0.31, p<0.01) and short tail (-0.34, p<0.01). Significant relationships were shown between progressive sperm motility and TZI (-0,56, p<0.01), proportions of normal sperm (0,67, p<0.01), elongated heads (-0,18, p<0.01), acrosome defects (-0,38, p<0.01), bent heads (-0,37, p<0.01), ERC (-0,23, p<0.01), coiled (-0,36, p<0.01) and short (-0,45, p<0.01) tails (S10 Table).

Percentage of pyriform head positively correlated with percentages of elongated, vacuolated heads, bent head and ERC (S11 Table).

## Discussion

The study revealed significant regional differences in DNA fragmentation level and incidences of sperm morphological defects in the residents of the cities with different air pollution levels. The residents of Kemerovo were characterized by elevated DNA fragmentation, poor sperm morphology and increased incidences of morphological defects of head (pyriform, elongated, round heads, abnormal acrosome and vacuolated chromatin), asymmetrical neck insertion and ERC. Previous study showed that men from the general population in Kemerovo had lowered sperm count, sperm concentration and progressive motility as well as serum testosterone and inhibin B level [56]. Regional variations of semen quality and sperm morphology are considered to be result of complex influences of various environmental conditions at the place of residence such as climate, trace elements in water or food, environmental pollution as well as lifestyle factors [51] and ethnic features associated with genetic background [60]. It should be noted that Novosibirsk and Kemerovo are located under the same climatic conditions and characterized by similar ethnicity of population and socio-cultural identity. However, Kemerovo is characterized by increased air pollution compared to Novosibirsk due to numerous chemical and coal industry enterprises [55, 56]. The industrial activities result in increased air emissions of solid particles, $SO_2$ nitrogen oxides in Kemerovo city. It should be noted that we investigated general urban populations affected by complex mixture of pollutants from industrial activity and transport (gases, PM, polycyclic hydrocarbons, heavy metals etc.) as well as negative lifestyle factors. Therefore, it is difficult to estimate the contribution of certain pollutants or other negative factors in spermiogenesis damage in investigated populations. In particular, our results suggest that smoking adversely affect spermiogenesis, resulting in decreased sperm count, sperm motility, elevated DFI, TZI and incidence of elongated, vacuolated heads, spermatozoa with ERC and asymmetrical neck insertion in smokers. The previous studies showed negative effects of smoking on sperm count, sperm motility and morphology and DNA fragmentation [61, 62]. These evidences suggest that smoking can contribute to impairment sperm morphology in men from Kemerovo and Novosibirsk. However, taking in to account lack of differences in frequencies of smoking, as well as other negative lifestyle factors (alcohol consumption, obesity), in Novosibirsk and Kemerovo, it is probably that

increased air pollution in Kemerovo is the main contributor of revealed regional differences in sperm morphology.

It was reported that air pollution influences adversely on sperm count, motility and morphology [44] as well as DNA fragmentation [63] and results in increasing incidence of sperm head defects [43, 64–68]. Our findings generally consistent with previous evidences and suggest that environmental air pollution affects sperm morphogenesis, with a distributions of sperm defects in previous and our investigations being specific. A prevalence of certain sperm defects probably reflect a nature of pollution and several other factors including pollution level, ethnicity, lifestyle etc. In our study environmental pollution of the territory was associated with increased incidence of elongated and pyriform heads and impaired acrosome. Pyriform and elongated heads is considered to be stress-induced abnormalities arising as a result of spermiogenesis errors due to negative external factors [22]. It was shown that varicocele [69, 70], severe psychological stress [71] scrotal hyperthermia [72] are associated with the increased incidence of elongated heads. At the ultrastructural level elongated and pyriform heads are characterized by an excessive nuclear elongation, a presence additional membranous layers between the outer and inner leaves of the nuclear envelope as well as a poorly condensed chromatin [35, 70, 73].

Several mechanisms has been proposed to explain effects of environmental pollution to semen quality, including oxidative stress [63, 74, 75], forming DNA adducts [44, 76] and hormonal imbalance [77, 78]. It was shown that the subjects from Kemerovo were characterized by somewhat lowered serum testosterone level and substantially decreased inhibin B level in comparison with men from Novosibirsk [56]. Lowered inhibin B level is associated with Sertoli cell and spermatogenic cell impairment [79, 80] and is a marker of a testicular damage due to toxic effects of pollutants [81, 82]. Therefore, lowered inhibin B level may indicate the disturbance of Sertoli cells and spermatogenesis in the subject from Kemerovo due to the industrial pollution. As shown by experimental studies in rodents, environmental pollution may directly affect Sertoli cells [44, 83, 84], resulting in Sertoli cell vacuolization, detachment and separation of germ cells from underlying epithelium, impairment of signaling pathways between Sertoli and germ cells, disruption of blood-testis-barrier and impairment of immunological microenvironment in the testis [85]. Furthermore, environmental toxicants induce a cytoskeletal disruption in Sertoli cells [86–88] and it may exert sperm morphology. It is known that sperm head shaping during spermiogenesis depend on a precise cooperation of cytoskeleton structures of spermatid and surrounding Sertoli cells. Ectoplasmic specializations of Sertoli cells are formed by layers of actin bundles and this structure reduces its diameter, exerting external forces to compact the spermatid nucleus during spermatid elongation. Moreover, acroplaxome and manchette are transition cytoskeleton structures which are formed during spermiogenesis [12]. The acroplaxome modulates the exogenous forces generated by Sertoli cells. The marginal ring of the acroplaxome and the perinuclear ring of the manchette reduce their diameter as they gradually descend along the nucleus toward the spermatid tail in a zipper-like movement that facilitates nuclear condensation and shaping [10]. Moreover, the acroplaxome anchors the developing acrosome to the nuclear envelope of spermatid [12]. Microtubule of the manchette are required for protein traffic from the spermatid nucleus to developing tail. This intramanchette transport is known to be necessary for head shaping and tail assembly [10, 89]. Rodent studies showed that the depletion of genes encoding microtubule-related proteins resulted in an excessive manchette and sperm head elongation [90], with sperm appearance resembling elongated and pyriform head in human. Taking into account these evidences it is reasonable to assume that excessive head elongation and disturbed acrosome formation may be caused by impaired function of manchette and acroplaxome in spermatids and/or cytoskeleton of Sertoli cells as well as motor or signal proteins related with its as a result of

**Table 6. Comparisons of DNA fragmentation levels measured by SCSA methods in subject from Eastern Siberia, European and Asian countries.**

| Reference | Country | Studied sample | Subject age, years | DFI, % |
|---|---|---|---|---|
| [91] | New Zeland | Subfertile men and semen donors (n = 1082) | 41(7.4) | 12.1(9.8) |
| [92] | Sweden | Military conscripts (n = 304) | 18.4(0.36) | 11.0(6.09) |
| [93] | Norway | Men recruited by advertisement (n = 199) | 29 (21–38) | 12.0(6.9) |
| [94] | USA | Fertile donors (n = 16) | No data | 15(11–21) |
| [95] | Sweden | Infertile patients (n = 350) | No data | 18.1(12.0) |
| [96] | UK | Subfertile patients (n = 56) | 36.9(5.4) | 22.8(14.6) |
| [97] | China | College students (n = 630) | 21.0 (21–22) | 10.4 (6.4–17.4)* |
| [97] | China | Reproductive medical center patients (n = 10362) | 33.0(30.0–38.0) | 12.3(8.1–18.1)* |
| **Our data** | **Russia, Novosibirsk** | **Men recruited by advertisement (n = 278)** | **22.2(3,51)** | **8.2(6,6)** |
| **Our data** | **Russia, Kemerovo** | **Men recruited by advertisement (n = 258)** | **23,7(5,0)** | **11,7(8,6)** |

Data are presented as Mean(SD) or Median (5th-95th percentiles).

disturbed microenvironment in testis due to environmental pollution affecting the studied population in Kemerovo. It should be noted that this hypothesis needs an experimental verification because the regulation of sperm head shaping is a complicated process requiring many proteins.

Our study showed that the residents of Kemerovo had elevated DNA fragmentation level in comparison with the men from Novosibirsk. As shown in Table 6, DFI obtained in our study for the residents of both Novosibirsk and Kemerovo were similar to its reported for fertile, subfertile subjects and the male from the general populations from European and Asian countries [91–99]. The mean values of DFI in the populations under our study may be considered as «low DNA fragmentation levels» [7, 59] and suggest that the observed increase of DFI in most men from Kemerovo city is quite moderate without severe negative effects on fertility. However, increased DNA fragmentation and percentage of vacuolated heads considering as indicators of poor chromatin integrity [98] suggest impaired chromatin condensation in the residents of Kemerovo.

Our study showed that men with lowered sperm motility and/or concentration characterized by poor sperm morphology and elevated incidence of most morphological defects. Increased DNA fragmentation level was associated with elevated TZI, incidence of abnormal acrosome, coiled and short tail as well as combined defects of heads midpiece and tail. The relationships between sperm morphology and sperm motility were reported [99]. Furthermore, it was shown that elevated DNA fragmentation was associated with poor sperm morphology and increased incidence of head and tail defects [24, 100–104], ROS production and impaired chromatin compaction [24]. The authors proposed that the revealed associations resulted from abnormal nuclear remodeling resulting in the abnormal sperm morphology, the increase of sperm ROS production as well as vulnerability of DNA to an oxidative attack due to poor chromatin condensation [24]. The results of our cross-sectional study supplement these evidences, showing that the geographic location affects the relationships between conventional semen parameters, sperm morphology and DNA fragmentations. DFI was closely related with sperm count and acrosome defects in men from Novosibirsk, but not Kemerovo. It should be noted that there are discrepancies in current literature on the relation of sperm morphology with other semen parameters and fertility because several studies reported it, whereas other found no differences for sperm morphology in men with different reproductive status [32–34] and DNA fragmentation level [30, 31]. The controversial evidences are considered to be caused by differences in recruitment method of the studied samples (proven fertility

men, infertility patients, men from general population etc) and/or sperm morphology assessment methodology including classification criteria, staining technique, investigator experience etc. [35]. In our study the recruitment protocol, the methods for estimating sperm count, sperm motility and DNA fragmentation, staining technique were the same, sperm morphology assessment was performed by the only experienced researcher, using standard classification criteria, the obtained data were corrected for age and abstinence time. Therefore, it is most likely that our evidences suggest that there are regional features in the manifestation of the relationships between sperm morphology pattern, sperm DNA fragmentation and conventional semen parameters. The reason of this phenomenon is unclear. It is possible that differences in external factors affecting population in certain regions can affect spermiogenesis in various ways, weakening or increasing the strength of the relationship between semen parameters. The present results suggest that a geographic location should be considered to understand the current evidences on morphology pattern as indicator impaired sperm function.

## Conclusion

The performed study showed significant regional variability in percentages of several sperm morphological defects and DNA fragmentation in the men from the general populations living in the cities with different levels of environmental pollution located in the Western Siberia. The residents of Kemerovo characterized by elevated DNA fragmentation, poor sperm morphology and increased incidences of pyriform, elongated, round heads, abnormal acrosome and vacuolated chromatin, asymmetrical neck insertion and ERC probably due to increased environmental pollution in Kemerovo. Moreover, the regional features in the relationships between sperm morphology, sperm count, motility and DNA fragmentation were shown, suggesting an importance of studying sperm morphology patterns in general populations of men from different regions.

## Supporting information

**S1 Table. The influence of sperm quality status on DFI and percentages of sperm morphology defects (ANCOVA results).** Bold text indicates significant ($p < 0.05$) influences showed by ANCOVA. SQS–sperm quality status: normozoospermia (sperm concentration and percentage of progressively motile spermatozoa equal to or above the lower reference limits) and pathozoospermia (sperm concentration < 15 mln/ml and/or percentage of progressively motile spermatozoa<32%). ERC–excess residual cytoplasm.
(DOCX)

**S2 Table. The relationship between semen quality and sperm morphology.** Results based on raw data. Analysis of variance used to compare all parameters. Significant ($p < 0.05$) differences between groups are highlighted by bold text. Abbreviations: SD—standard deviation; (5–95) - 5th–95th percentile; DFI–DNA fragmentation index; TZI–teratozoospermia index; ERC–excess residual cytoplasm.
(DOCX)

**S3 Table. The influence of DNA fragmentation level on sperm quality and percentage of sperm morphology defects (ANCOVA results).** Significant ($p < 0.05$) effects of factors are highlighted by bold text. Abbreviations: DFI–DNA fragmentation index; TZI–teratozoospermia index; ERC–excess residual cytoplasm.
(DOCX)

**S4 Table. The effects of smoking on sperm quality and sperm morphology (ANCOVA results).** Significant ($p < 0.05$) effects of factors are highlighted by bold text. Abbreviations:

DFI–DNA fragmentation index; TZI–teratozoospermia index; ERC–excess residual cytoplasm.
(DOCX)

**S5 Table. The effects of alcohol consumption on sperm quality and sperm morphology (ANCOVA results).** Significant (p<0.05) effects of factors are highlighted by bold text. Abbreviations: DFI–DNA fragmentation index; TZI–teratozoospermia index; ERC–excess residual cytoplasm.
(DOCX)

**S6 Table. The effects of alcohol consumption on sperm quality and sperm morphology.** Results based on raw data. Analysis of variance used to compare all parameters. Significant (p<0.05) differences between groups are highlighted by bold text. Abbreviations: SD—standard deviation; (5–95) - 5th–95th percentile; DFI–DNA fragmentation index; TZI–teratozoospermia index; ERC–excess residual cytoplasm.
(DOCX)

**S7 Table. The effects of obesity on sperm quality and sperm morphology (ANCOVA results).** Significant (p<0.05) effects of factors are highlighted by bold text. Abbreviations: DFI–DNA fragmentation index; TZI–teratozoospermia index; ERC–excess residual cytoplasm.
(DOCX)

**S8 Table. The effects of obesity on sperm quality and sperm morphology.** Results based on raw data. Analysis of variance used to compare all parameters. Significant (p<0.05) differences between groups are highlighted by bold text. Abbreviations: SD—standard deviation; (5–95) - 5th–95th percentile; DFI–DNA fragmentation index; TZI–teratozoospermia index; ERC–excess residual cytoplasm.
(DOCX)

**S9 Table. Correlations DFI with other semen parameters and sperm morphology.** Bold text indicates statistically significant (p<0.05) correlation coefficients. ERC–excess residual cytoplasm.
(DOCX)

**S10 Table. Spearman's correlation between percentages of sperm morphology defects and other semen parameters.** Bold text indicates significant (p<0.05) correlation coefficients. ERC–excess residual cytoplasm.
(DOCX)

**S11 Table. Spearman's correlation coefficients among percentages of different sperm morphology defects.** Bold text indicates significant (p<0.05) correlation coefficients. DH–double head; BH–bent head; ERC–excess residual cytoplasm; ANI–asymmetrical neck insertion; DT–double tail; CT–coiled tail; ST–short tail.
(DOCX)

## Acknowledgments

We thank physicians Andrei Erkovich, Natalia Voroschilova, Natalia Kuznezova, for coordinating the recruitment and performing a physical examination of participants, Ms. Natalia Gutorova for help in collection of questionnaires, Ms. Daria Tataru for estimating DNA fragmentation. All the volunteers participating in the study are thanked.

## Author Contributions

**Conceptualization:** Alexander Osadchuk, Ludmila Osadchuk.

**Data curation:** Alexander Osadchuk, Ludmila Osadchuk.

**Funding acquisition:** Ludmila Osadchuk.

**Investigation:** Maxim Kleshchev, Alexander Osadchuk.

**Methodology:** Maxim Kleshchev.

**Project administration:** Ludmila Osadchuk.

**Supervision:** Alexander Osadchuk.

**Writing – original draft:** Maxim Kleshchev.

**Writing – review & editing:** Alexander Osadchuk, Ludmila Osadchuk.

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
