## [Decision Letter · Decision Letter 0]

23 Jul 2021

PONE-D-21-18446

Impaired semen quality, an increase of sperm morphological defects and DNA fragmentation associated with environmental pollution in urban population of young men from Western Siberia, Russia

PLOS ONE

Dear Dr. Kleshchev,

Thank you for submitting your manuscript to PLOS ONE. After careful consideration, we feel that it has merit but does not fully meet PLOS ONE’s publication criteria as it currently stands. Therefore, we invite you to submit a revised version of the manuscript that addresses the points raised during the review process.

The study is very interesting and provides important information in relation to environmental pollution and sperm quality. The rationale and conclusions will be further strengthened by providing additional information on the exact pollution parameters in the two cities, co-morbidities of the analyzed population, life style habits (smoking, alcohol consumption, medication, etc) and rigorous statistical analyses. Correlations, if any, between the sperm quality and other parameters (life style) should be carefully analyzed.

We look forward to receiving your revised manuscript.

Kind regards,

Suresh Yenugu

Academic Editor

PLOS ONE

Journal Requirements:

Reviewers' comments:

Reviewer's Responses to Questions

**Comments to the Author**

1. Is the manuscript technically sound, and do the data support the conclusions?

Reviewer #1: Partly

Reviewer #2: Yes

2. Has the statistical analysis been performed appropriately and rigorously? 

Reviewer #1: Yes

Reviewer #2: Yes

3. Have the authors made all data underlying the findings in their manuscript fully available?

Reviewer #1: Yes

Reviewer #2: No

4. Is the manuscript presented in an intelligible fashion and written in standard English?

Reviewer #1: Yes

Reviewer #2: Yes

5. Review Comments to the Author

Reviewer #1: This study discusses two cities with the same climate in different regions of Russia. Because of the different environmental pollution, the researchers compared the sperm quality data and DNA fragmentaion index of men in the two places. Researchers have found that in cities with high environmental pollution, abnormal sperm types, sperm quality, and sperm DNA fragmentation index will indeed deteriorate. The researchers attributed this observation to the results of different levels of pollution in the two cities.

The advantages of this research report are: It includes sperm morphology examination, DNA fragmentation examination is very complete, and the number of cases is sufficient, which is worthy of appreciation.

Disadvantages of this article:

1. No direct evidence related to sperm quality and environmental pollution has been provided. The researchers explained the two phenomena directly related to each other. Although the possibility is very high, as a scientific report, there should be objective evidence and related explanations. The explanation of relevance should be the most difficult part of this research. If it is not possible to directly explain the relationship between sperm type and environmental pollution, it is recommended that the author only needs to make an objective observation and analysis of the observation results of the two places.

2. Researchers should provide actual observational data on environmental pollution in the two places.

3. The reliability of sperm morphology observation will be inaccurate based on the examiner's subjective bias. Whether the inspections in the two cities have been verified. The author can provide a way of verification.

Reviewer #2: The authors reported on ‘Impaired semen quality, an increase of sperm morphological defects and DNA fragmentation associated with environmental pollution in urban population of young

men from Western Siberia, Russia’ with the male volunteers in Novosibirsk (n=278) and Kemerovo (n=258) were assessed for their environmental pollution impact on sperm morphology and sperm DNA fragmentation level. Specifically, we commend the author`s effort to investigate the sperm morphology and sperm DNA fragmentation (not sperm concentration or motility) associated with environmental pollution because there is lack of this topic. This point has a value on this article.

The authors have concluded that The residents of Kemerovo characterized by elevated DNA fragmentation, poor sperm morphology and increased incidences of pyriform, elongated, round heads, abnormal acrosome and vacuolated chromatin, asymmetrical neck insertion and ERC probably due to increased environmental pollution in Kemerovo.

I have a few comments:

Comment #1: The author suggested that spermiogenesis is sensitive to environmental conditions and lifestyle factors such as obesity, smoking, andrological disorders and chronic system illness that affect sperm morphology in introduction section. In study population, precise inclusion, exclusion criteria is mandatory (Varicocele, cryptorchidism, past history of genitalia surgery etc.).

In Table 1, adding detail demographic information such as smoking, DM, any infertile disease would be beneficial.

Comment #2: The two cities are located to same climatic condition

Please mention detail climatic condition of the city (Detail level of particulate matter). Are there any other confounding factors other than climatic condition which could cause detrimental impact on sperm morphology and sperm DNA fragmentation?

Kemerivi city was characterized by increased environmental pollution especially by particulate matter due to coal mining and the numerous metallurgical and chemical industries.

Compare the detail information coal mining and the numerous metallurgical and chemical industries between two cities.

Comment #3: Line 155 genial tract -> genital tract , Table 1 Heght ->Height

Mention abbreviations in all Tables, and describe exact number of P-values in Tables as well.

6. PLOS authors have the option to publish the peer review history of their article (what does this mean?). If published, this will include your full peer review and any attached files.

Reviewer #1: No

Reviewer #2: **Yes: **Dae Keun Kim

---

## [Author Response · Author response to Decision Letter 0]

14 Sep 2021

The authors thank the editor and reviewers for a detailed critical analysis of the manuscript. We have carefully analyzed your comments and revised our manuscript according to them. We would like to answer your questions and give some explanations. All changes made to the manuscript are highlighted in red color.

According to the recommendations of the editor and reviewers we added the information about pollution parameters in Novosibirsk and Kemerovo cities, frequency of negative life style factors (smoking, alcohol consumption, obesity) and andrological disorders. Moreover, we investigated effects of these negative factors on sperm morphology and DFI. The results obtained presented and discussed in the revisited manuscript. 

Response to reviewer #1:

1. No direct evidence related to sperm quality and environmental pollution has been provided. The researchers explained the two phenomena directly related to each other. Although the possibility is very high, as a scientific report, there should be objective evidence and related explanations. The explanation of relevance should be the most difficult part of this research. If it is not possible to directly explain the relationship between sperm type and environmental pollution, it is recommended that the author only needs to make an objective observation and analysis of the observation results of the two places.

According to the recommendation we added to the manuscript the information on pollution levels in Novosibirsk and Kemerovo (Table 2 in the revisited manuscript). The data were obtained from public government reports about environment status and conservation of Kemerovo [http://kuzbasseco.ru/doklady/o-sostoyanii-okruzhayushhej-sredy-kemerovskoj-oblasti/] and Novosibirsk [https://www.nso.ru/page/2624] regions, where it presented as emission volumes of pollutants. As shown in Table 2, Kemerovo was characterized by severe increased emissions volume in atmospheric air of solid particles, sulfur dioxide, nitrogen oxides and carbon monoxide in comparison with Novosibirsk city. Moreover, we provided the additional information about the life style factors (smoking, alcohol consumption, obesity, current andrological disorders) which could affect sperm morphology. It was shown that smoking adversely affected spermiogenesis, resulting in decreased sperm count, sperm motility, elevated DFI, TZI and incidence of elongated, vacuolated heads, spermatozoa with ERC and asymmetrical neck insertion in smokers compared to non-smokers. Alcohol consumption, obesity and presence of andrological disorders did not affect DFI and percentages of most sperm defects. Taking in to account lack of differences in frequencies of smoking, as well as other negative lifestyle factors (alcohol consumption, obesity), in Novosibirsk and Kemerovo, it is most likely that increased air pollution in Kemerovo is the main contributor of revealed regional differences in sperm morphology.

2. Researchers should provide actual observational data on environmental pollution in the two places.

According to the your recommendation we added to the manuscript the information on pollution levels in Novosibirsk and Kemerovo (Table 2 in the revisited manuscript).

3. The reliability of sperm morphology observation will be inaccurate based on the examiner;s subjective bias. Whether the inspections in the two cities have been verified. The author can provide a way of verification.

We agree with the distinguished reviewer that quality control is necessary for accurate sperm morphology assessment. To perform quality control, percentage of normal sperm was estimated twice in random and blinded order by two independent experienced investigators. Spearman correlation coefficient between percentages sperm with normal morphology obtained by first and second count was 0.78, statistically significant difference was not observed. The differences between first and second counts of normal sperm morphology were acceptable according quality control criteria provided by «WHO laboratory manual…» for each sample. The mean value for percentage of normal sperm was calculated. 

Unfortunately, due to labor intensity percentages of sperm morphological defects were estimated once. However Counting sperm morphology defects was performed by the only experienced researcher.in random and blinded order. The slides were coded, randomly mixed and the examiner did not know about origin and other related characteristics of the sample (geographic locations, DFI level etc.). Random and blinded order lets to avoid effects of subjective bias during sperm morphology assessment and used in population studies of sperm morphology (Auger et al., 2001). We described quality control procedure mentioned above in the section "Materials and methods" of the manuscript.

Response to reviewer #2

1. The author suggested that spermiogenesis is sensitive to environmental conditions and lifestyle factors such as obesity, smoking, andrological disorders and chronic system illness that affect sperm morphology in introduction section. In study population, precise inclusion, exclusion criteria is mandatory (Varicocele, cryptorchidism, past history of genitalia surgery etc.). In Table 1, adding detail demographic information such as smoking, DM, any infertile disease would be beneficial. 

Inclusion criteria for participation in the study were absence of acute general diseases or chronic illness in an acute phase, and genital tract infections. The participants with urogenital disorders were included to examine the population more fully.

According to the your recommendations, of the manuscript we added detail information in the table 1 about occurrence of the life style factors (smoking, alcohol consumption, obesity, current andrological disorders) which could affect sperm morphology in the populations in Novosibirsk and Kemerovo. 

The two cities are located to same climatic condition Please mention detail climatic condition of the city (Detail level of particulate matter). Are there any other confounding factors other than climatic condition which could cause detrimental impact on sperm morphology and sperm DNA fragmentation? 

Kemerivi city was characterized by increased environmental pollution especially by particulate matter due to coal mining and the numerous metallurgical and chemical industries. Compare the detail information coal mining and the numerous metallurgical and chemical industries between two cities.

According to the recommendation we added to the manuscript the information on pollution levels in Novosibirsk and Kemerovo (Table 2 in the revisited manuscript). The data were obtained from public government reports about environment status and conservation of Kemerovo [http://kuzbasseco.ru/doklady/o-sostoyanii-okruzhayushhej-sredy-kemerovskoj-oblasti/] and Novosibirsk [https://www.nso.ru/page/2624] regions. As shown in Table 2, Kemerovo city was characterized by severe increased emissions volume in atmospheric air of solid particles, sulfur dioxide, nitrogen oxides and carbon monoxide in comparison with Novosibirsk city. Average annual temperature do not differ in these cities.

To estimate effects of confounding factors, we investigated the influence of lifestyle factors (cigarette smoking, alcohol consumption, obesity) which could affect spermiogenesis in men from studied population on DFI and sperm morphology. It was shown that smoking adversely affected spermiogenesis, resulting in decreased sperm count, sperm motility, elevated DFI, TZI and incidence of elongated, vacuolated heads, spermatozoa with ERC and asymmetrical neck insertion in smokers. Alcohol consumption, obesity and presence of andrological disorders did not affect percentages of most sperm defects. Taking in to account lack of differences in frequencies of smoking, as well as other negative lifestyle factors (alcohol consumption, obesity), in Novosibirsk and Kemerovo, it is probably that increased air pollution in Kemerovo is the main contributor of revealed regional differences in sperm morphology.

3. Line 155 genial tract -> genital tract , Table 1 Heght ->Height Mention abbreviations in all Tables, and describe exact number of P-values in Tables as well.

All mentioned and other misprints corrected. P -values and decoding of the abbreviations in the tables are presented in the revisited manuscript.

References.

1. World Health Organization. Laboratory manual for the examination and processing of human semen. 5th ed. Geneva: WHO; 2010.

2. Auger J, Eustache F, Andersen AG, Irvine DS, Jørgensen N, Skakkebaek NE, et al. Sperm morphological defects related to environment, lifestyle and medical history of 1001 male partners of pregnant women from four European cities. Hum Reprod 2001;16: 2710-7.

---

## [Decision Letter · Decision Letter 1]

8 Oct 2021

Impaired semen quality, an increase of sperm morphological defects and DNA fragmentation associated with environmental pollution in urban population of young men from Western Siberia, Russia

PONE-D-21-18446R1

Dear Dr. Kleshchev,

We’re pleased to inform you that your manuscript has been judged scientifically suitable for publication and will be formally accepted for publication once it meets all outstanding technical requirements.

Kind regards,

Suresh Yenugu

Academic Editor

PLOS ONE

Additional Editor Comments (optional):

Reviewers' comments:

Reviewer's Responses to Questions

**Comments to the Author**

1. If the authors have adequately addressed your comments raised in a previous round of review and you feel that this manuscript is now acceptable for publication, you may indicate that here to bypass the “Comments to the Author” section, enter your conflict of interest statement in the “Confidential to Editor” section, and submit your "Accept" recommendation.

Reviewer #2: All comments have been addressed

2. Is the manuscript technically sound, and do the data support the conclusions?

Reviewer #2: Yes

3. Has the statistical analysis been performed appropriately and rigorously? 

Reviewer #2: Yes

4. Have the authors made all data underlying the findings in their manuscript fully available?

Reviewer #2: Yes

5. Is the manuscript presented in an intelligible fashion and written in standard English?

Reviewer #2: Yes

6. Review Comments to the Author

Reviewer #2: The authors have adequately addressed the comments in a previous round of review process.

There is no issues about dual publication, and research ethics.

7. PLOS authors have the option to publish the peer review history of their article (what does this mean?). If published, this will include your full peer review and any attached files.

Reviewer #2: **Yes: **Dae Keun Kim, e-mail: kdg070723@gmail.com

---

## [Editor Report · Acceptance letter]

13 Oct 2021

PONE-D-21-18446R1 

Impaired semen quality, an increase of sperm morphological defects and DNA fragmentation associated with environmental pollution in urban population of young men from Western Siberia, Russia 

Dear Dr. Kleshchev:

I'm pleased to inform you that your manuscript has been deemed suitable for publication in PLOS ONE. Congratulations! Your manuscript is now with our production department. 

Kind regards, 

on behalf of

Dr. Suresh Yenugu 

Academic Editor

PLOS ONE